# Inhibiting Transglutaminase 2 Mediates Kidney Fibrosis via Anti-Apoptosis

**DOI:** 10.3390/biomedicines10061345

**Published:** 2022-06-07

**Authors:** Jong-Joo Moon, Yejin Choi, Kyu-Hyeon Kim, Areum Seo, Soie Kwon, Yong-Chul Kim, Dong-Ki Kim, Yon-Su Kim, Seung-Hee Yang

**Affiliations:** 1Biomedical Research Institute, Seoul National University, Seoul 03080, Korea; jongjoomoon114@gmail.com (J.-J.M.); yejstar98@gmail.com (Y.C.); kxk584@case.edu (K.-H.K.); sal4113@hanmail.net (A.S.); 2Department of Internal Medicine, Seoul National University Hospital, Seoul 03080, Korea; soie1127@naver.com (S.K.); imyongkim@gmail.com (Y.-C.K.); dkkim73@gmail.com (D.-K.K.); yonsukim@snu.ac.kr (Y.-S.K.); 3Department of Internal Medicine, Seoul National University College of Medicine, Seoul 03080, Korea; 4Kidney Research Institute, Seoul National University Medical Research Center, Seoul 03080, Korea

**Keywords:** transglutaminase 2, chronic kidney disease, cystamine, apoptosis, fibrosis

## Abstract

Transglutaminase 2 (TG2) is a calcium-dependent transamidating acyltransferase enzyme of the protein-glutamine γ-glutamyltransferase family implicated in kidney injury. In this study, we identified associations between TG2 and chronic kidney disease (CKD) identified by visualizing TG2 in kidney biopsy samples derived from CKD patients using immunohistochemistry and measuring the plasma TG2 concentrations. Our study revealed a connection between TG2 and the pathological markers of kidney disease. We showed high plasma TG2 levels in samples from patients with advanced CKD. In addition, we observed an increase in TG2 expression in tissues concomitant with advanced CKD in human samples. Moreover, we investigated the effect of TG2 inhibition on kidney injury using cystamine, a well-known competitive inhibitor of TG2. TG2 inhibition reduced apoptosis and accumulation of extracellular molecules (ECM) such as fibronectin and pro-inflammatory cytokine IL-8. Collectively, the increased expression of TG2 that was observed in advanced CKD, hence inhibiting TG2 activity, could protect kidney cells from ECM molecule accumulation, apoptosis, and inflammatory responses, thereby preventing kidney fibrosis.

## 1. Introduction

Chronic kidney disease (CKD) is manifested as a progressive deterioration of the kidney function that results from various complicated responses, such as metabolic changes, apoptosis, oxidative stress, and inflammation. The incidence of CKD disease worldwide has steadily increased over the past decades [1]. More importantly, CKD not only affects renal health but is also associated with other health hazards like increased cardiovascular risk, renal replacement therapy, and shortened life expectancy [2]. Known risk factors that might cause CKD are diabetes mellitus, hypertension, and hyperlipidemia. One of the most distinguishable pathological features of CKD is kidney fibrosis, which occurs as a result of excessive extracellular matrix (ECM) accumulation [3].

Transglutaminase 2 (TG2) is a ubiquitously distributed enzyme that catalyzes the transamidation of proteins in a calcium-dependent manner and regulates various biological processes, such as ECM accumulation, tumor progression, cell differentiation, and cell death. Various studies have demonstrated its role in several diseases, including neurodegenerative diseases [4], autoimmune disorders [5], and tissue fibrosis in various organs [6,7]. For example, kidney tissue derived from a subnephrected rat model showed increased TG2 expression accompanied by fibrotic changes [8].

Increased extracellular TG2 is closely associated with fibrosis as it forms stable ε-(γ-glutamyl)-lysine crosslinks by catalyzing an acyl-transfer reaction (EC 2.3.2.13) between the ε-amino group of peptide-bound lysine and the γ-carboxamide group of glutamine [9]. Following its secretion from the cell, TG2 catalyzes extensive crosslinking of ECM, making it resistant to degradation by proteases and thereby leading to accelerated ECM deposition [10].

Apoptosis is one of the mechanisms underlying kidney injury and is implicated in TG2 activity. In CKD, apoptosis is steadily increased and induces tubular epithelial cell depletion, tubular atrophy, and loss of kidney function [11]. TG2 may exhibit either a pro-apoptotic or anti-apoptotic effect depending on the condition of the cell [12]. An increased cytosolic Ca^2+^ concentration above a threshold leads to the inactivation of the transamidation activity of TG2 by GTP [12]. This sensitizes the cells to apoptosis and facilitates the apoptotic process. Several studies have demonstrated that excess reactive oxygen species increase cytosolic Ca^2+^, leading to intracellular TG2 activation, which can result in oxidative stress-induced apoptosis [13].

Chronic inflammation induces injury of kidney cells in the CKD environment. Inflammation is orchestrated by several cytokines, chemokines, and their receptors. Elevated TG2 activity is interconnected to pro-inflammatory cytokines such as tumor necrosis factor α, nuclear factor-kappa B (NF-κB), mitogen-activated protein kinases, and several kinds of interleukins [14,15,16]. In the kidney, the activated pro-inflammatory signaling pathways recruit immune cells and induce inflammation and accumulation of ECM components, which eventually leads to renal cell death [17,18]. The inhibition of TG2 is a potential therapeutic strategy for treating various diseases because of the association between TG2 and pathologic activity.

TG2 overexpression has been observed in several kidney disease animal models. In a 5/6 nephrectomy rat model that represents progressive kidney fibrosis, the mRNA and protein expressions of TG2 were observed to be increased. At enhanced levels, TG2 interacts with the heparan sulphate family, particularly syndecan-4, which activates TGFβ1 related ECM accumulation [19]. In a kidney transplant animal model, activated monocytes demonstrated TG2 overexpression accompanied with increased levels of the apoptotic marker caspase 3 [20]. Another animal model showed increased urine and serum TG2 concentrations that accompanied kidney disease progression [21].

Cystamine is an irreversible allosteric inhibitor of TG2, and it is reduced to cysteamine in vivo, a competitive inhibitor of the transamidation activity of TG2 [22]. TG2 suppression using cystamine has been reported to be associated with reduced collagen 1 formation in the kidney [23]. Another TG2 inhibitor, cysteamine, also reportedly reduces fibrosis by reducing oxidative stress and myofibroblast activity in unilateral ureteral obstruction models [24]. The pharmacological suppression of TG2 has been reported to decrease apoptosis in the brain and vascular smooth muscle cells [25,26] and reduce fibrosis and progression in CKD and diabetic kidney animal models [27,28].

Considering these relationships between TG2 and its pathologic activities that cause kidney injury, we hypothesized that patients with progressed CKD and deteriorated kidney function demonstrate higher TG2 expression in their serum and kidney tissues. We also hypothesized that TG2 inhibition induces renoprotective effects demonstrated via the suppression of inflammation, apoptosis, and ECM accumulation. However, the effect of cystamine on the inflammatory pathway and apoptosis in primary cultured human kidney tubular epithelial cells (hTECs) has not been evaluated. To validate our hypotheses, we assessed TG2 expressions in kidney tissues and plasma derived from patients with CKD and analyzed their kidney function markers and performed TG2 inhibition using cystamine, and evaluated the effect on fibrotic, apoptotic, and inflammatory markers in hTECS.

## 2. Materials and Methods

### 2.1. Unilateral Ureteral Obstruction (UUO) Animal Model

All animal experiments were performed by following the Seoul National University Hospital Institutional Animal Care and Use Committee-approved protocols (permit number: 20-0009-S1A0). Seven-week-old male C57BL/6 mice (20 g) were used for this study. Mice were purchased from Koatech (Seoul, Korea). Anesthesia was performed using intraperitoneal xylazine (Rompun; 10 mg/kg of body weight; Bayer, Mississauga, ON, Canada), followed by administration of tiletamine mixed with zolazepam (1:1) (ZoletilTM; 30 mg/kg of body weight; Virbac, Carroll, France). A flank incision was made to approach the left kidney. The ureter ligation was conducted using 5-0 silk at two points, and then cut between the ligatures. The mice were placed on a heating pad (37 °C), and intraperitoneal administration of pre-warmed phosphate buffered saline (PBS; 37 °C) was performed to prevent dehydration during the procedure. The same procedures, except for ureter ligation, were conducted for mock mice. Mice were sacrificed at 3, 7, and 14 days after ureter ligation and then, UUO kidneys were harvested for evaluation.

### 2.2. 5/6 Nephrectomy (5/6 Nx.) Animal Model

All animal experiments were performed following the Seoul National University Hospital Institutional Animal Care and Use Committee-approved protocols (permit number: 20-0072-S1A0). The 5/6 nephrectomy rat model was adopted to examine the change in TG2 expression in the animal CKD model. Main branches of the left renal artery of Sprague Dawley rats (250 g, 6~7 weeks old; Koatech, Seoul, Korea) were ligated. After 7 days, right nephrectomy was conducted for removal of 5/6 of kidney mass; 4 or 8 weeks following model establishment, animals were euthanized; then, kidneys were harvested and examined.

### 2.3. Terminal Deoxynucleotidyl Transferase dUTP Nick-End Labeling (TUNEL) Assay

For the detection of apoptosis in the UUO and 5/6 Nx. animal model, we conducted a TUNEL (Roche, Mannheim, Germany) assay in accordance with the manufacturer’s instructions. To visualize TUNEL positive nuclei, 4′-6-diamidino-2-phenylindole (DAPI, Invitrogen, Carlsbad, CA, USA) was used. To evaluate TUNEL positive nuclei, at least five fields were randomly selected and captured using a Leica TCS SP8 STED CW instrument (20/0.7 numerical aperture objective lens of a DMI 6000 inverted microscope; Leica, Mannheim, Germany) and Meta-Morph software (version 7.8.10).

### 2.4. Human Samples

Kidney biopsy cores and plasma samples from IgA nephropathy (IgAN), DM nephropathy (DMN), and normal kidneys were collected at Seoul National University Hospital from 2013 to 2019 under protocols that were approved by the Institutional Review Board. Kidney diseases, including IgAN and DMN, and control groups (CTL) were distinguished using pathological diagnosis based on renal biopsy. The control group was defined based on pathological non-specific findings. Patient data collected at the time of kidney biopsy using electronic medical records included age, sex, body mass index, estimated glomerular filtration rate (eGFR), urine protein/creatinine ratio, serum hemoglobin, serum albumin, and creatinine.

### 2.5. Quantitative Real-Time PCR

Total RNA isolated from kidney tissues of unilateral ureteral obstruction (UUO) mice 3, 7, and 14 days after UUO model establishment was subjected to real-time PCR to analyze TG2 mRNA levels. RNeasy mini kit^®^ (Qiagen, Hilden, Germany) was used for total RNA extraction. Reverse transcription was performed using oligo-dT primers and AMV-RT Taq polymerase (Promega, Madison, WI, USA). Real-time PCR was carried out on an ABI 7500 thermocycler (Applied Biosystems, Foster City, CA, USA) using the Assay-on-Demand SYBR Green method. GAPDH mRNA levels were used for normalization. The sequences of the forward and reverse primers used in the study were as follows: GAPDH forward: 5′-TATGTCGTGGAGTCTACTGGT-3′, GAPDH reverse: 5′-GAGTTGTCATATTTCTCGT-3′, TG2 forward: 5-′GGACATCACCCATACCTACAAG-3′, TG2 reverse: 5′-TCTCTGCCAGTTTGTTCAGG-3′.

### 2.6. Histopathological Evaluations

A nephropathologist used light, electron, and immunofluorescence microscopy for the assessment of human kidney biopsy cores. Kidney tissues were stained with hematoxylin and eosin for light microscopy evaluation. For each sample, at least five fields (magnification: ×100), were randomly selected and then evaluated. The severity of the tubular pathology was evaluated. Tubular pathology, including tubular atrophy, interstitial fibrosis, and inflammatory cell infiltration, was classified into 3 classes based on the affected area (normal: 0, mild: ≤25%, moderate: 26–50%, marked >50%, or diffuse pattern) [29].

### 2.7. Immunohistochemical Evaluation of TG2

Kidney tissues were cut into 4-μm paraffin sections and stained with Sirius red (Abcam, Cambridge, UK) to observe their histology. Deparaffinization and hydration were performed using xylene and ethanol, respectively. To block endogenous streptavidin activity, 3% hydrogen peroxide (H_2_O_2_, Sigma-Aldrich, St. Louis, MO, USA) was used. The deparaffinized sections were stained with anti-mouse TG2 antibody (Novus Biologicals, Centennial, CO, USA). These sections were then incubated with secondary antibodies, goat anti-rabbit (Vector Laboratories, Burlingame, CA, USA) and rabbit anti-rat IgG (Vector Laboratories, Burlingame, CA, USA). The sections were then counterstained with Mayer’s hematoxylin (Sigma-Aldrich) and visualized under a light microscope (DFC-295; Leica, Mannheim, Germany). For rodent and human kidney samples, at least five fields (magnification: ×100) were randomly selected, and Sirius Red and brown-stained areas were quantified using computer-based morphometric analysis (Qwin 3; Leica).

### 2.8. Enzyme-Linked Immunosorbent Assay (ELISA)

The concentrations of TG2 in kidney lysate and cell culture media were evaluated using a commercial ELISA kit (R&D Systems, Minneapolis, MN, USA).

### 2.9. Human Primary Tubular Epithelial Cell (hTEC) Preparation

The protocol for obtaining and processing human kidney specimens was reviewed and approved by the Institutional Review Board of Seoul National University Hospital (IRB No. 2110-026-1260). We dissected the proximal renal tubular segments from kidneys of patients diagnosed with renal cell carcinoma [30]. After dissecting the cortex, the unaffected specimens were minced and digested with Hank’s balanced salt solution (HBSS) containing 1.5 mg/mL collagenase (Sigma-Aldrich) at 37 °C for 1 h. The digested kidney sample was then washed through a series of sieves (150, 120, 70, and 40 μm) with 1 × PBS and centrifuged at 500× *g* for 5 min. After recovering cells from the pellet, they were incubated in DMEM/F12 (Lonza, Basel, Switzerland) for 4 h. Tubules floating in the media were collected and cultured on collagen-coated Petri dishes (BD Biosciences, Franklin Lakes, NJ, USA) until colonies of epithelial cells formed; 2–3 passages of these epithelial cells were used in the current study. We added 2 ng/mL of recombinant transforming growth factor β (rTGFβ) and cultured for 48 h to induce cellular injury including fibrotic change and apoptosis. Various concentrations of cystamine were administrated simultaneously with rTGFβ in the treatment group.

### 2.10. Cell Apoptosis Assay

To induce apoptosis in the primary cultures of hTECs, 2 ng/mL of recombinant TGFβ (R&D Systems) was used. After 48 h of rTGFβ administration, we subjected the hTECs to flow cytometry (BD Biosciences) using Annexin V/propidium iodide (PI) fluorescein isothiocyanate (FITC) apoptosis kit (BD Biosciences), to evaluate cellular apoptosis and necrosis. The harvested cells (5 × 10^5^) were washed with cold PBS and resuspended in 100 μL binding buffer. The cells were then stained with 5 μL FITC conjugated Annexin V (10 mg/mL) and 10 μL PI (50 mg/mL). The stained cells were incubated for half an hour at room temperature in the dark. BD FACSDiva (version 8.0; BD Biosciences) was used to acquire and analyze the flow cytometry data.

### 2.11. Western Blot Analysis

hTECs were collected 48 h after rTGFβ induction. Protein was extracted using radioimmunoprecipitation assay buffer containing Halt protease inhibitor (Pierce, Rockford, IL, USA). Extracted proteins were aliquoted in equal amounts (30–60 μg) and then loaded on 10% sodium dodecyl sulfate polyacrylamide gels for electrophoresis. The proteins were then transferred onto Immobilon FL 0.4 μM polyvinylidene difluoride membranes (Millipore, Bedford, MA, USA) that were probed with fibronectin (Santa Cruz, Dallas, TX, USA), E-cadherin (Abcam), and β-actin (Sigma-Aldrich). Probed proteins were detected using a chemiluminescence system (ECLTM PRN 2106; Amersham Pharmacia Biotech, Buckinghamshire, UK) and a gel documentation system (Gel Doc 1000 and Multi-Analyst version 1.1, Bio-Rad Laboratories, Hercules, CA, USA). The images of the western blots were semi-quantified using ImageJ software (ImageJ Ver. 1.52a, Wayne Rasband, National Institutes of Health, Bethesda, MD, USA).

### 2.12. Statistical Analysis

Statistical analysis was conducted using GraphPad Prism 9 (GraphPad Software, San Diego, CA, USA). Student’s *t*-test was used for comparison. For categorical variables, the chi-square test was used to compare two groups. One-way analysis of variance using Tukey’s test was used to compare more than two groups. The mean and standard errors of the mean were used to present the results. Statistical significance was set at *p* < 0.05.

## 3. Results

We used the UUO mouse model and 5/6 Nx. model to determine the association between histopathological changes and TG2 levels in renal tissues. Kidneys from UUO mice were harvested 3, 7, and 14 days after UUO model establishment. To assess the degree of fibrosis, Sirius Red staining was performed. The Sirius red positive area, which reflects collagen deposition, showed a time-dependent increase (CTL vs. 3 days vs. 7 days vs. 14 days; 2.53 ± 0.45% vs. 11.23 ± 1.11% vs. 24.13 ± 2.36% vs. 30.04 ± 3.04%, Figure 1a,b, left panel). The fibrotic kidney tissue demonstrated elevated levels of TG2 (4.11 ± 0.56% vs. 17.45 ± 0.99% vs. 23.38 ± 1.87% vs. 23.57 ± 1.86%, Figure 1b, right panel) and a significant increase of TG2 mRNA was observed in the kidney with aggravated pathological changes (0.99 ± 0.10 vs. 4.17 ± 0.55 vs. 5.71 ± 0.92 vs. 8.98 ± 1.84, fold, Figure 1c). Apoptosis in UUO kidney showed a continuous increasing over time (0.61 ± 0.16% vs. 3.21 ± 0.69% vs. 8.18 ± 1.32% vs. 58.45 ± 6.54%, Figure 1a,d). TG2 tissue expression was evaluated at 4 and 8 weeks after the model establishment for 5/6 Nx. model. Similar to the UUO model, Sirius red positive area (CTL vs. 4 weeks vs. 8 weeks; 0.58 ± 0.08% vs. 20.92 ± 0.94% vs. 27.792 ± 1.11%, Figure 1e,f, left panel) and tissue-expressed TG2 levels (0.33 ± 0.08% vs. 20.58 ± 0.95% vs. 24.78 ± 1.18%, Figure 1e,f, right panel) were significantly increased after model establishment. The proportion of TUNEL positive apoptosis was also constantly increased at 4 and 8 weeks compared with that of the control group in 5/6 Nx. (0.70 ± 0.23% vs. 16.55 ± 2.17% vs. 77.72 ± 4.90%, Figure 1e,g).

### 3.1. TG2 Levels Are Increased with Deterioration of the Kidney in Human Samples

The baseline characteristics of patients whose plasma TG2 concentrations were measured are presented in Table 1. The IgAN group demonstrated no significant differences in BMI, serum albumin, serum creatinine, eGFR, and plasma levels compared with those of the DMN group. However, patients with IgAN were younger, had higher blood hemoglobin, and demonstrated lesser urine protein/creatinine ratio when compared to the patients with DMN (Table 1).

Patients with advanced CKD showed higher levels of plasma TG2 concentrations (CKD 1, 2 vs. CKD 3 vs. CKD 4, 5; 1876 ± 60.98 vs. 2044 ± 38.24 vs. 2126 ± 67.55, in pg/mL; Figure 2a, left panel). An elevated plasma TG2 concentration was observed in samples from patients with higher proteinuria (urine protein/creatinine ratio >1 vs. urine protein/creatinine ratio <1; 1920 ± 43.89 vs. 2063 ± 41.16, pg/mL; Figure 2a, right panel).

Immunohistochemical staining for TG2 was performed to assess the presence of TG2 in kidney biopsies from patients with CKD. Baseline patient characteristics of the biopsy core are represented in Table 2. Tissue donors without CKD were used as the control group. No significant differences were observed in age, BMI, blood hemoglobin, serum creatinine, and eGFR between the IgAN and the DMN groups, although the DMN group showed relatively lower albumin level, smaller TG2 positive area, and more proteinuria (Table 2).

In the early stage of IgAN, the tissue expression of TG2 in IgAN was mainly located in the tubule and the interstitial area with minor expression in the glomeruli. Contrarily, in early DMN, TG2 expression was higher than that of IgAN in the glomeruli and was also observed in relatively smaller tubular and interstitial areas. Advanced IgAN showed increased glomerular TG2 expression compared with that in early IgAN, and TG2 expression in advanced DMN was accompanied by tubular and interstitial expression. Heightened TG2 expression was observed in patients with decreased eGFR; it was the highest in the CKD 4, 5 group (3.39 ± 0.54% vs. 6.89 ± 0.86% vs. 8.55 ± 1.02% vs. 12.89 ± 1.11%, Figure 2b,c, left panel). Patients with proteinuria (more than 1 g/day) demonstrated elevated TG2 expression (6.21 ± 0.69% vs. 10.02 ± 0.83%, Figure 2c middle panel). TG2 expression was inversely related to the eGFR (Figure 2d, *p* < 0.0001, R2 = 0.295). Patients with worsened tubular conditions, including tubular atrophy, tubular fibrosis, and inflammatory cell infiltration, showed increased TG2 (Figure 2e).

We observed a definite correlation with increased TG2 expression, kidney function markers, and pathological findings. Collectively, our data clearly showed increased TG2 tissue expression in advanced CKD.

### 3.2. TG2 Inhibition Using Cystamine Decreased rTGFβ-Induced Apoptosis and Suppressed Fibrotic Changes in Primary Cultures of Tubular Epithelial Cells

To determine the effect of TG2 inhibition on rTGFβ-induced fibrosis, we conducted western blot analysis for fibronectin and E-cadherin (Figure 3a). Cystamine-mediated TG2 inhibition led to reduced fibronectin (rTGFβ vs. rTGFβ + cystamine 0.5 mM vs. rTGFβ + cystamine 1 mM vs. rTGFβ + cystamine 2 mM vs. rTGFβ + cystamine 4 mM; 10.84 ± 1.47 vs. 12.07 ± 1.69 vs. 8.97 ± 1.16 vs. 6.23 ± 0.75 vs. 2.28 ± 0.28 fold, Figure 3b, left panel) and increased E-cadherin (rTGFβ vs. rTGFβ + cystamine 0.5 mM vs. rTGFβ + cystamine 1 mM vs. rTGFβ + cystamine 2 mM vs. rTGFβ + cystamine 4 mM; 0.81 ± 0.01 vs. 0.92 ± 0.16 vs. 1.58 ± 0.24 vs. 1.55 ± 0.22 vs. 1.19 ± 0.16 fold Figure 3b, right panel). The concentration of the pro-inflammatory cytokine IL-8 in cell culture media was measured to determine the effect of TG2 inhibition on inflammation. Administration of cystamine alleviated IL-8 concentration (rTGFβ vs. rTGFβ + cystamine 0.5 mM vs. rTGFβ + cystamine 1 mM vs. rTGFβ + cystamine 2 mM vs. rTGFβ + cystamine 4 mM; 12.4.8 ± 9.36 pg/mL vs. 166.6 ± 10.21 pg/mL vs. 106.4 ± 7.25 pg/mL vs. 73.64 ± 8.77 pg/mL vs. 79.76 ± 9.53 pg/mL, Figure 3c).

To determine the effect of TG2 inhibition on apoptosis in renal tubular cells, we treated tubular epithelial cells with rTGFβ. After 48 h of rTGFβ treatment, apoptosis was evaluated using annexin V/PI assay. hTECs that were stimulated with rTGFβ showed increased apoptosis compared with control groups (CTL vs. rTGFβ; 5.35 ± 0.46% vs. 11.97 ± 0.22%, *p* < 0.0001, Figure 3d). Cystamine-stimulated TECs exhibited decreased apoptosis compared to rTGFβ-treated cells (rTGFβ vs. rTGFβ + cystamine 0.5 mM vs. rTGFβ + cystamine 1 mM vs. rTGFβ + cystamine 2 mM; 11.97 ± 0.22% vs. 7.58 ± 0.26% vs. 8.63 ± 0.58% vs. 7.03 ± 0.34%, Figure 3d).

## 4. Discussion

The present study demonstrated elevation of TG2 expression in the plasma and kidney tissues in advanced CKD compared with early CKD. The inhibition of TG2 activity resulted in decreased apoptosis and attenuation of the release of IL-8 from rTGFβ-stimulated hTECs. Intriguingly, our immunohistochemistry results showed distinguishable tissue TG2 expression patterns between IgAN and DMN, especially in early CKD. Differences in TG2 expression patterns could be clarified through clinical features of both glomerulopathies. Clinically, IgAN showed more hematuria and less proteinuria compared with DMN in the early stage. A large amount of proteinuria is more likely to be induced by the glomerulus than tubules; in this sense, it is possible to explain the higher TG2 expression in the glomeruli of DMN than IgAN. Our results that patients with more than 1 g/day of proteinuria demonstrate higher tissue TG2 expression and plasma TG2 levels also support this. However, despite the differences in the causes of CKD and pattern of TG2 expression, increased immunoreactivity of TG2 was accompanied by worsening of kidney function markers.

Although TG2 is prevalent in the cytosol, it is also distributed in the nucleus, mitochondria, plasma membrane, and ECM. TG2 is extremely relevant to wound healing via fibroproliferation. However, TG2 overexpression is associated with fibrosis, leading to reduced functions in various organs. In an animal model for heart failure, the hypertrophied ventricle tissue demonstrated upregulated TG2 [31]. Similarly, TG2-overexpressing hearts showed decreased cardiac function, such as depressed ejection fraction, cardiac contractility, and increased diffuse interstitial fibrosis of the myocardium [32,33]. In the lung, increased TG2 expression was observed in lung fibrotic tissue of idiopathic pulmonary fibrosis patients [34]. Additionally, liver specimens with fibrosis showed elevated TG2 levels in the extracellular space, and a strong statistical relationship between TG2 expression and fibrosis stage has been reported [35].

CKD is attributed to a complex array of physiological phenomena such as metabolic changes, apoptosis, oxidative stress, and inflammation, causing irreversible damage to the kidneys, thereby compromising kidney function. Herein, we used rTGFβ to induce apoptosis and fibrosis and showed that cystamine-induced suppression of TG2 in rTGFβ-treated primary cultures of hTECs had significant anti-apoptotic and anti-fibrotic effects. TGFβ plays a major role in kidney fibrosis by inducing the proliferation of myofibroblasts and accumulation of ECM components [36]. Higher levels of TG2 activate the inactive form TGFβ. Activated TGFβ crosslinks latent TGFβ1 binding protein with ECM components such as fibronectin on the surface of the cell [37,38] and induces ECM expansion by increasing the synthesis of ECM components [39]. It also inhibits ECM degradation by suppressing the synthesis of metalloproteinases [40]. TG2 catalyzes the cross-linking of integrins and fibronectins, which confers resistance to proteolysis. Additionally, TG2 inhibition results in diminished deposition of ECM and facilitates the turnover of ECM components [41,42,43].

Suppressing TG2 is associated with decreased apoptosis and prolonged survival. However, suppressing TG2 does not always protect cells from apoptosis. Several types of malignant cells with enhanced TG2 expression show decreased apoptosis when treated with doxorubicin and radiation [44,45]. TG2 induces caspase-independent apoptosis by forming crosslinks and inhibiting Sp1, which enhances the expression of growth factor receptors required for cell survival, including c-Met and EGF receptors. It also triggers the mitochondrial release of apoptosis-inducing factor and cytochrome c, both of which lead to apoptosis [46,47]. Furthermore, TG2-induced apoptosis is associated with the retinoblastoma protein. TG2 inactivates the Rb protein and inhibits its interaction with E2F-1 transcription factor, enhancing E2F-1 degradation and accelerating apoptosis [48]. The association of TG2 with apoptosis corroborates the results obtained in this study (Figure 3d).

TG2 regulates the expression of pro-inflammatory cytokines, including IL-8, in the NF-κB signaling pathway [49]. TG2 activates NF-κB by promoting the degradation of NF-κB inhibitor (IKK) or stimulating the phosphorylation of the RelA/p65 NF-κB subunit [50]. NF-κB activation may aid in macrophage survival and trigger continuous activation of the inflammatory process [51]. Hence, inhibiting TG2 is a promising way to deal with inflammation and kidney damage. Previous reports have demonstrated elevated levels of IL-8 in advanced CKD [52]. Increased IL-8 is associated with higher mortality and dialysis requirements, which is observed in patients with CKD who have higher IL-8 concentrations than the normal population [53]. In chronic inflammatory conditions such as CKD, IL-8 promotes the activation and recruitment of immune cells, primarily neutrophils. These stimulated neutrophils release proteases that intensify damage to the kidney [54]. A previous study has reported that increased IL-8 triggers apoptosis via its interaction with chemokine receptor 1 [55]. This could partially explain our observation of decreased IL-8 and apoptosis in cystamine-mediated TG2 suppressed cells (Figure 3c).

Increased urinary TG2 levels depict allograft inflammation or fibrosis in kidney transplant patients [56], and in CKD patients, urinary TG2 levels could be considered a potential biomarker for CKD progression [21]. Previous studies have demonstrated the mechanisms underlying TG2 associated-ECM accumulation. TECs excrete TG2 in exosome to interstitial ECM through interaction with syndecan-4, a proteoglycan belonging to the heparan sulfate family [9,10]. In the ECM, TG2 contributes to the crosslinking with integrins and fibronectins that concede resistance to proteolysis, and TG2 inhibition resulted in diminished deposition and facilitated the turnover of ECM components [41,42,43]. Due to the pathologic activities associated with TG2 which result in kidney injury, several investigations have been conducted in this area. Genetic TG2 inhibition in mouse models including UUO and LPS injection exhibited reduced ECM molecule accumulation, decreased TGFβ1 levels [57], and suppressed infiltration of neutrophils in the kidney [58]. There have been few clinical trials on TG2 inhibitors. A monoclonal Ab for TG2, zampilimab, has been investigated in phase I/II trials for chronic allograft rejections of transplanted kidneys [59]. Another TG2 inhibitor, ZED1227, ameliorated gluten-induced duodenal mucosal damage in patients with celiac disease. Schuppan et al. have suggested a mechanism that ZED1227 exerts a protective effect by blocking the formation of deamidated gluten and gluten-specific CD4+ T cell activation in the duodenum [60]. However, further investigations are needed to explore the clinical implications of TG2 inhibitors.

This study has several limitations. We demonstrated the anti-apoptotic, anti-fibrotic, and anti-inflammatory effects of inhibiting TG2 only in an in vitro environment. Moreover, the present study revealed only the protective phenomenon caused by TG2 inhibition, and did not include experiments to ascertain the exact mechanisms underlying these protective effects. Further studies using various methods such as multi-omics analysis or in animal experiments are required to investigate the detailed mechanism and the effect of TG2 inhibition in vivo.

## 5. Conclusions

Collectively, our work depicts that TG2 expression was increased in various advanced CKD samples. Inhibiting TG2 activity could protect kidney cells from ECM component accumulation, apoptosis, and inflammatory responses that lead to kidney injury. With respect to the clinical implications, our study signifies that measuring TG2 expressions could facilitate the evaluation of the severity of CKD progression and presents the detailed beneficial phenomenon that results from pharmacologic TG2 inhibition in kidney TECs.

## Figures and Tables

**Figure 1 biomedicines-10-01345-f001:**
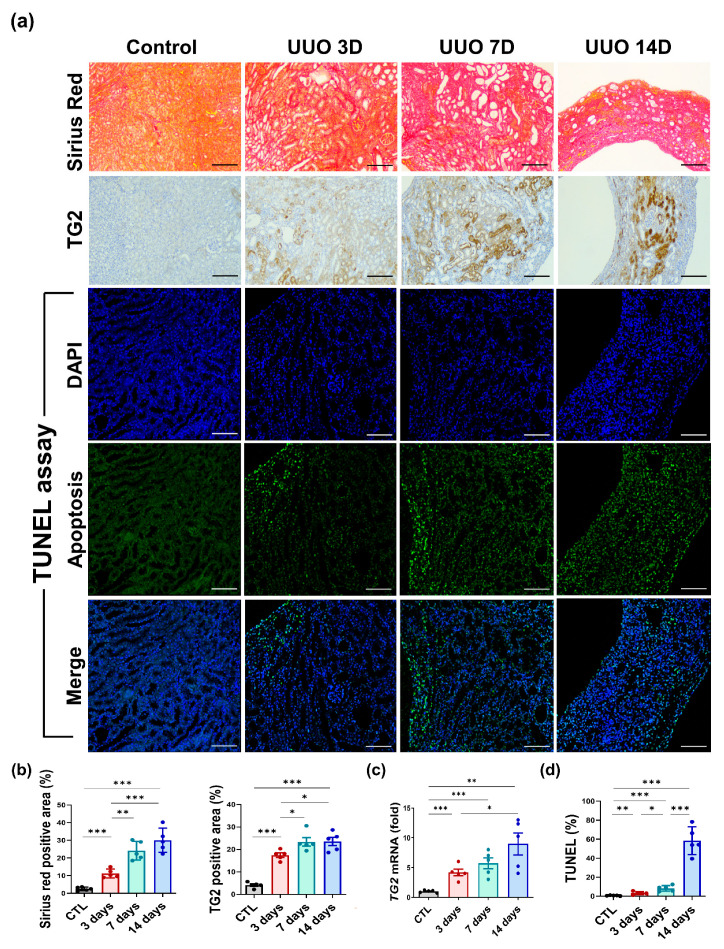
Increased TG2 levels were accompanied by pathological aggravation in UUO and 5/6 nephrected (Nx.) rodent models. UUO kidneys were evaluated thrice, on days 3, 7, and 14, after the establishment of the UUO model. (**a**) Representative images of UUO kidneys stained with Sirius red, immunohistochemical staining for TG2, and TUNEL assay on days 3, 7, and 14. (**b**) Semi-quantification results for Sirius red and TG2-stained UUO kidney. (**c**) Changes in TG2 mRNA abundance after 3, 7, and 14 days from UUO surgery. (**d**) Percentage of TUNEL positive area on days 3, 7, and 14. (**e**) In the 5/6 Nx. model, kidney tissue-expressed TG2 levels were evaluated twice at 4 and 8 weeks. Representative images of 5/6 Nx. kidneys stained with Sirius red, immunostained with TG2, and TUNEL assay. (**f**) Semi-quantification results for TG2-stained 5/6 Nx. Kidney. (**g**) Apoptosis was assessed twice at 4 and 8 weeks using the TUNEL assay. (Scale bar = 100μm, * *p* <0.05, ** *p* < 0.01, *** *p* < 0.001).

**Figure 2 biomedicines-10-01345-f002:**
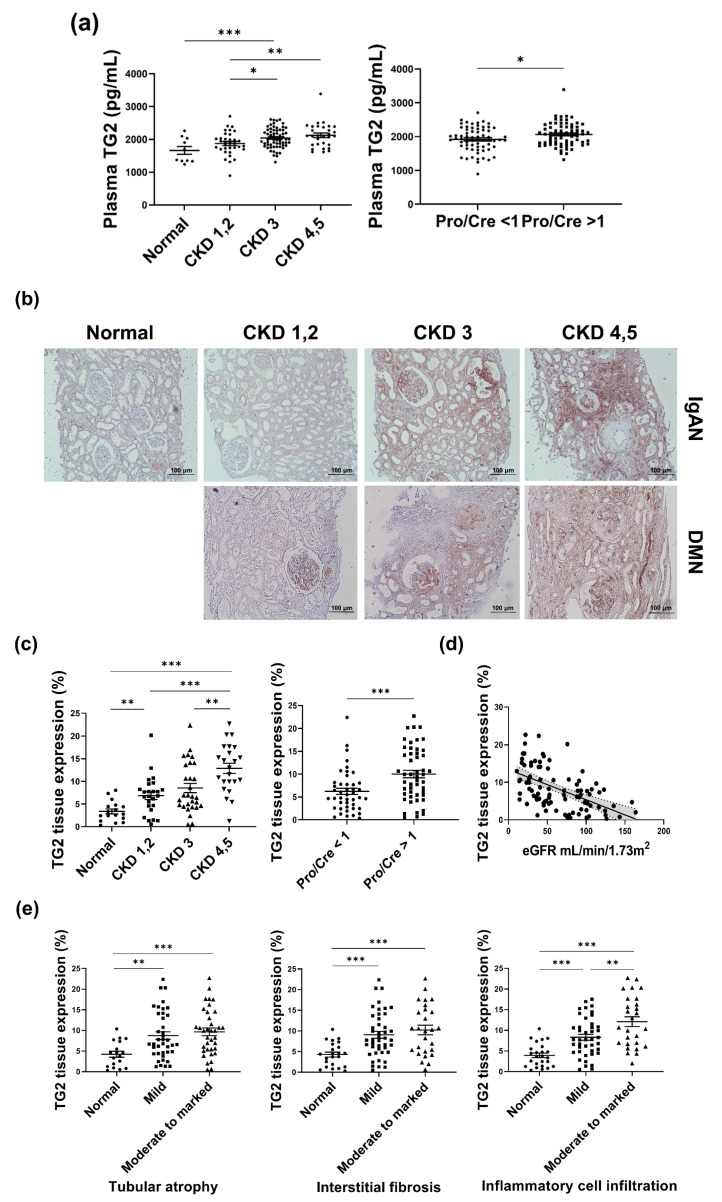
TG2 levels were increased with deterioration of kidney function and pathological aggravation in patient samples. (**a**) The concentration of TG2 was evaluated in the plasma of CKD patients. (**b**) Representative images of immunohistochemical staining for TG2. (**c**) Differences in the tissue expression level of TG2 according to kidney function markers, including CKD stage and urine protein/creatinine ratio. (**d**) Association between TG2 expression and eGFR. (**e**) Differences in TG2 expression levels according to tubular pathology (* *p* < 0.05, ** *p* < 0.01, *** *p* < 0.001).

**Figure 3 biomedicines-10-01345-f003:**
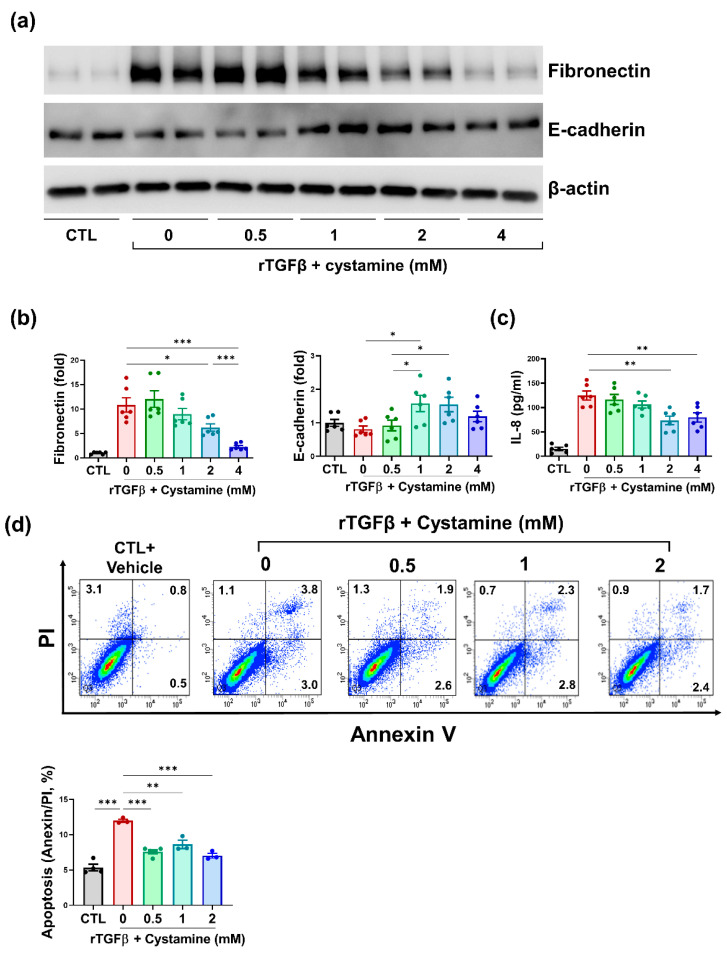
Cystamine-mediated inhibition of TG2 protected hTECs from rTGFβ-induced injury. (**a**) Representative images of western blots for fibronectin and E-cadherin. (**b**) Semi-quantification results for fibronectin and E-cadherin. (**c**) Concentration of IL-8 in cell culture media. (**d**) Evaluation for cell death conducted using Annexin V/PI assay in hTECs 48 h after rTGFβ induction (* *p* < 0.05, ** *p* < 0.01, *** *p* < 0.001).

**Table 1 biomedicines-10-01345-t001:** Patient characteristics (plasma samples).

Variable	Total (*n* = 137)	IgAN (*n* = 100)	DMN (*n* = 27)	CTL (*n* = 10)	*p* Value			
					Group	IgAN vs. CTL	DMN vs. CTL	IgAN vs. DMN
Age (year)	46.88 ± 15.73	46.09 ± 15.47	53.44 ± 14.06	42.00 ± 5.99	0.041	0.495	0.053	0.019
Sex (male, %)	81 (59.12)	58 (58)	18 (66.66)	5 (50)	0.597			
BMI (kg/m^2^)	24.12 ± 3.56	24.11 ± 3.60	24.51 ± 3.232	23.16 ± 4.11	0.593	0.436	0.301	0.597
CKD stage 1, 2 (%)	34 (24.82)	27 (27)	7 (25.93)					
CKD stage 3 (%)	63 (45.99)	50 (50)	13 (48.15)					
CKD stage 4, 5 (%)	30 (21.90)	23 (23)	7 (25.93)					
Blood hemoglobin (g/dL)	12.26 ± 2.05	12.41 ± 1.98	11.19 ± 1.95	13.69 ± 1.86	0.005	0.052	0.001	0.005
Serum albumin (g/dL)	3.84 ± 0.54	3.85 ± 0.56	3.69 ± 0.50	4.07 ± 0.39	0.145	0.236	0.039	0.176
Creatinine (mg/dL)	1.76 ± 1.53	1.79 ± 1.59	2.04 ± 1.50	0.78 ± 0.19	0.169	0.048	0.013	0.467
eGFR (CKD-EPI, mL/min/1.73 m^2^)	56.44 ± 33.79	54.08 ± 32.90	47.50 ± 26.42	104.2 ± 22.26	<0.001	<0.001	<0.001	0.340
Urine protein/creatinine ratio	1.78 ± 2.06	1.55 ± 2.01	3.06 ± 2.08	0.60 ± 0.28	0.002	0.140	0.027	0.001
Plasma TG2 level (pg/mL)	1990 ± 357.8	2030 ± 347.8	1963 ± 334.0	1665 ± 381.3	0.021	0.002	0.001	0.371

Values are means ± SD for continuous variables and *n* (%) for categorical variables. BMI: Body Mass Index.

**Table 2 biomedicines-10-01345-t002:** Characteristics of patients with TG2 expression in kidney tissue.

Variable	Total (*n* = 94)	IgAN (*n* = 56)	DMN (*n* = 21)	CTL (*n* = 17)	*p*-Value			
					Group	IgAN vs. CTL	DMN vs. CTL	IgAN vs. DMN
Age (year)	44.95 ± 17.90	45.84 ± 15.16	47.24 ± 11.26	39.18 ± 17.90	0.205	0.133	0.099	0.702
Sex (male, %)	56 (56.57)	36 (64.29)	14 (66.67)	6 (35.29)	0.077			
BMI (kg/m^2^)	23.75 ± 3.72	24.28 ± 3.12	25.04 ± 3.97	25.22 ± 4.00	0.694	0.514	0.927	0.476
CKD stage 1, 2 (%)		20 (35.71)	5 (23.81)					
CKD stage 3 (%)		20 (35.71)	9 (42.86)					
CKD stage 4, 5 (%)		16 (28.57)	7 (33.33)					
Blood hemoglobin (g/dL)	12.26 ± 2.10	12.24 ± 2.00	11.44 ± 2.37	13.34 ± 1.68	0.019	0.043	0.008	0.141
Serum albumin (g/dL)	3.81 ± 0.59	3.85 ± 0.49	3.42 ± 0.77	4.14 ± 0.34	<0.001	0.031	0.001	0.004
Creatinine (mg/dL)	1.56 ± 1.00	1.66 ± 0.99	1.95 ± 1.10	0.75 ± 0.17	<0.001	<0.001	<0.001	0.259
eGFR (CKD-EPI, mL/min/1.73 m^2^)	64.26 ± 37.75	56.49 ± 33.39	50.51 ± 36.81	106.8 ± 19.39	<0.001	<0.001	<0.001	0.498
Urine protein/creatinine ratio	2.42 ± 3.42	1.75 ± 2.17	5.77 ± 5.01	0.47 ± 0.30	<0.001	0.019	<0.001	<0.001
TG2 positive area (%)	8.24 ± 5.61	10.26 ± 5.25	6.76 ± 5.72	3.39 ± 2.24	<0.001	<0.001	0.028	0.013

Values are means ± SD for continuous variables and *n* (%) for categorical variables. BMI: Body Mass Index.

## Data Availability

The data presented in this study are available in the main article.

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
