# Peer review of "Inhibiting Transglutaminase 2 Mediates Kidney Fibrosis via Anti-Apoptosis"

_biomedicines, 2022, doi:10.3390/biomedicines10061345_

Round 1

Reviewer 1 Report

This manuscript by Moon et al. investigates the relationship between expression of transglutaminase 2 (TG2) and chronic kidney disease (CKD). These are my comments:

  1. Please add a hypothesis statement at the end of the introduction and more information on what is known about TG2 in kidney disease.
  2. A statement of an Institutional IACUC approving the animal studies is missing in the manuscript. Please add.
  3. Please clarify in the methods section how long after UUO kidneys were collected.
  4. A detailed description of the cystamine treatment is missing in the methods section. Please include manufacturer, dose, length of treatment and administration route.
  5. Please also add the histological stains that were used for evaluation of kidney damage in human samples. How many fields/sample were evaluated and at which magnification?
  6. Similarly, please also describe in the methods section how the inflammatory cell infiltration depicted in Figure 2e was evaluated. How many fields were quantified and what markers were used for this quantification?
  7. The size of the histological photos included in Figure 1 is too small to distinguish the differences between the groups, including the localization of TG2. Please provide bigger panels for the reader to be able to appreciate.
  8. What comparison does the p value in tables 1 and 2 refer to? Which of the 3 groups are being compared: IgAN, DMN or normal? Please clarify in the manuscript and table.
  9. Figure 2c – I believe that the authors forgot to add the asterisks indicating that the CKD4,5 group is significantly greater than the normal group. Please revise.
  10. In line 276, the authors mention that there was a decreased TG2 activity in cells treated with cystamine and they refer the reader to Figure 3d; however, these data is not shown. Please revise and add these missing data.
  11. The authors show a decreased level of apoptosis in hTECs treated with cystamine, however, they do not show evidence of increased apoptosis in the animal models or human samples used in these studies. Did they find differences in TUNEL staining in these samples, for instance?
  12. Please make sure that all acronyms are defined the first time that they are used in the text.
  13. Please elaborate in the discussion on the limitations of the current study and what are the clinical perspectives of the findings here described. Are any TG2 inhibitors approved for use in humans?

Author Response

Reviewer 1

This manuscript by Moon et al. investigates the relationship between expression of transglutaminase 2 (TG2) and chronic kidney disease (CKD). These are my comments:

  1. Please add a hypothesis statement at the end of the introduction and more information on what is known about TG2 in kidney disease.

à According to the reviewer’s suggestion, we have added information on TG2 in kidney disease and a hypothesis statement at the end of the introduction section.

Introduction

TG2 overexpression has been observed in several kidney disease animal models. In a 5/6 nephrectomy rat model that represents progressive kidney fibrosis, the mRNA and protein expressions of TG2 were observed to be increased. At enhanced levels, TG2 inter-acts with the heparan sulphate family, particularly syndecan-4, which activates TGFβ1 related ECM accumulation [19]. In a kidney transplant animal model, activated monocytes demonstrated TG2 overexpression accompanied with increased levels of the apoptotic marker caspase 3 [20]. Another animal model showed increased urine and serum TG2 concentrations that accompanied kidney disease progression [21].

Considering these relationships between TG2 and its pathologic activities that cause kidney injury, we hypothesized that patients with progressed CKD and deteriorated kidney function demonstrate higher TG2 expression in their serum and kidney tissues. We also hypothesized that TG2 inhibition induces renoprotective effects demonstrated via the suppression of inflammation, apoptosis, and ECM accumulation. However, the effect of cystamine on the inflammatory pathway and apoptosis in primary cultured human kid-ney tubular epithelial cells (hTECs) has not been evaluated. To validate our hypotheses, we assessed TG2 expressions in kidney tissues and plasma derived from patients with CKD and analyzed their kidney function markers and performed TG2 inhibition using cystamine and evaluated the effect on fibrotic, apoptotic, and inflammatory markers in hTECS.

  1. A statement of an Institutional IACUC approving the animal studies is missing in the manuscript. Please add.

à We have added institutional IACUC statements pertaining to the approval of animal studies, in the methods section.

2.1 Unilateral ureteral obstruction (UUO) animal model

All animal experiments were performed following the Seoul National University Hospital Institutional Animal Care and Use Committee-approved protocols (permit numbers: 20-0009-S1A0).

2.2 5/6 Nephrectomy (5/6 Nx.) animal model

All animal experiments were performed following the Seoul National University Hospital Institutional Animal Care and Use Committee-approved protocols (permit numbers: 20-0072-S1A0).

  1. Please clarify in the methods section how long after UUO kidneys were collected.

à As per the reviewer’s suggestion, we have added the relevant explanation at the end of the methods section describing the UUO animal model.

Mice were sacrificed at 3, 7, and 14 days after ureter ligation, then UUO kidneys were harvested for evaluation.

  1. A detailed description of the cystamine treatment is missing in the methods section. Please include manufacturer, dose, length of treatment and administration route.

à According to the reviewer’s comment, we have added information on TGFβ induction at the end of the section describing human primary tubular epithelial cell (hTEC) preparation.

We added 2 ng/mL of recombinant transforming growth factor β (rTGFβ) and cultured for 48 h to induce cellular injury including fibrotic change and apoptosis. Various concentrations of cystamine were administrated simultaneously with rTGFβ in the treatment group.

  1. Please also add the histological stains that were used for evaluation of kidney damage in human samples. How many fields/sample were evaluated and at which magnification?

à As a response to the reviewer’s comment, we have added the relevant explanation in the methods section (2.6) describing the immunohistochemical evaluation of TG2.

For rodent and human kidney samples, at least five fields (magnification: × 100) were randomly selected, and Sirius Red and brown-stained areas were quantified using computer-based morphometric analysis (Qwin 3; Leica).

  1. Similarly, please also describe in the methods section how the inflammatory cell infiltration depicted in Figure 2e was evaluated. How many fields were quantified and what markers were used for this quantification?

à According to the reviewer’s comment, we have added the relevant explanation in the methods section (2.5) describing histopathological evaluations. In our study, a nephropathologist determined the degree of inflammatory cell infiltration using HE stained biopsy core.

Kidney tissues were stained with Hematoxylin and Eosin for light microscopy evaluation. For each sample, at least five fields (magnification: X100), were randomly selected then evaluated.

  1. The size of the histological photos included in Figure 1 is too small to distinguish the differences between the groups, including the localization of TG2. Please provide bigger panels or the reader to be able to appreciate.

à We have revised Figure 1 to a larger sized figure according to your suggestion. 

  1. What comparison does the p value in tables 1 and 2 refer to? Which of the 3 groups are being compared: IgAN, DMN or normal? Please clarify in the manuscript and table.

à According to your suggestion, we have revised Table 1 and Table 2. We have also added the description pertaining to the table in the results section.

Table 1. Patient characteristics (plasma samples)

Variable

Total

(n = 137)

IgAN

(n = 100)

DMN

(n = 27)

CTL

(n =  10)

P value

Group

IgAN vs. CTL

DMN vs. CTL

IgAN vs. DMN

Age (year)

46.88±15.73

46.09±15.47

53.44±14.06

42.00±5.99

0.041

0.495

0.053

0.019

Sex (male, %)

81 (59.12)

58 (58)

18 (66.66)

5 (50)

0.597

BMI (kg/m2)

24.12±3.56

24.11±3.60

24.51±3.232

23.16±4.11

0.593

0.436

0.301

0.597

CKD stage 1,2 (%)

34 (24.82)

27 (27)

7 (25.93)

CKD stage 3 (%)

63 (45.99)

50 (50)

13 (48.15)

CKD stage 4,5 (%)

30 (21.90)

23 (23)

7 (25.93)

Blood hemoglobin (g/dL)

12.26±2.05

12.41±1.98

11.19±1.95

13.69±1.86

0.005

0.052

0.001

0.005

Serum albumin (g/dL)

3.84±0.54

3.85±0.56

3.69±0.50

4.07±0.39

0.145

0.236

0.039

0.176

Creatinine (mg/dL)

1.76±1.53

1.79±1.59

2.04±1.50

0.78±0.19

0.169

0.048

0.013

0.467

eGFR (CKD-EPI, mL/min/1.73m2)

56.44±33.79

54.08±32.90

47.50±26.42

104.2±22.26

<0.001

<0.001

<0.001

0.340

Urine protein/creatinine ratio        

1.78±2.06

1.55±2.01

3.06±2.08

0.60±0.28

0.002

0.140

0.027

0.001

Plasma TG2 level (pg/mL)

1990±357.8

2030±347.8

1963±334.0

1665±381.3

0.021

0.002

0.001

0.371

Values are means ± SD for continuous variables and n (%) for categorical variables. BMI: Body Mass Index.

Result

The baseline characteristics of patients whose plasma TG2 concentrations were measured are presented in Table 1. The IgAN group demonstrated no significant differences in BMI, serum albumin, serum creatinine, eGFR, and plasma levels compared with those of the DMN group. However, patients with IgAN were younger, had higher blood hemoglobin, and demonstrated lesser urine protein/creatinine ratio when compared to the patients with DMN (Table 1).

Variable

Total

(n = 94)

IgAN

(n = 56)

DMN

(n = 21)

CTL

(n =  17)

P value

Group

IgAN vs. CTL

DMN vs. CTL

IgAN vs. DMN

Age (year)

44.95±17.90

45.84±15.16

47.24±11.26

39.18±17.90

0.205

0.133

0.099

0.702

Sex (male, %)

56 (56.57)

36 (64.29)

14 (66.67)

6 (35.29)

0.077

BMI (kg/m2)

23.75±3.72

24.28±3.12

25.04±3.97

25.22±4.00

0.694

0.514

0.927

0.476

CKD stage 1,2 (%)

20 (35.71)

5 (23.81)

CKD stage 3 (%)

20 (35.71)

9 (42.86)

CKD stage 4,5 (%)

16 (28.57)

7 (33.33)

Blood hemoglobin (g/dL)

12.26±2.10

12.24±2.00

11.44±2.37

13.34±1.68

0.019

0.043

0.008

0.141

Serum albumin (g/dL)

3.81±0.59

3.85±0.49

3.42±0.77

4.14±0.34

<0.001

0.031

0.001

0.004

Creatinine (mg/dL)

1.56±1.00

1.66±0.99

1.95±1.10

0.75±0.17

<0.001

<0.001

<0.001

0.259

eGFR (CKD-EPI, mL/min/1.73m2)

64.26±37.75

56.49±33.39

50.51±36.81

106.8±19.39

<0.001

<0.001

<0.001

0.498

Urine protein/creatinine ratio

2.42±3.42

1.75±2.17

5.77±5.01

0.47±0.30

<0.001

0.019

<0.001

<0.001

TG2 positive area (%)

8.24±5.61

10.26±5.25

6.76±5.72

3.39±2.24

<0.001

<0.001

0.028

0.013

Table 2. Characteristics of patients with TG2 expression in kidney tissue

Values are means ± SD for continuous variables and n (%) for categorical variables. BMI: Body Mass Index

Results

Immunohistochemical staining for TG2 was done to assess the presence of TG2 in kidney biopsies from patients with CKD. Baseline patient characteristics of the biopsy core are represented in Table 2. Tissue donors without CKD were used as the control group. No significant differences were observed in age, BMI, blood hemoglobin, serum creatinine, and eGFR between the IgAN and the DMN groups, although the DMN group showed rela-tively lower albumin level, smaller TG2 positive area, and more proteinuria (Table 2).

  1. Figure 2c – I believe that the authors forgot to add the asterisks indicating that the CKD4,5 group is significantly greater than the normal group. Please revise.

à We have revised Figure 2c according to the reviewer’s comment and have included asterisks to show the statistical differences.

  1. In line 276, the authors mention that there was a decreased TG2 activity in cells treated with cystamine and they refer the reader to Figure 3d; however, these data is not shown. Please revise and add these missing data.

à Unfortunately, we did not perform experiments to measure TG2 activity. Our former manuscript caused a misunderstanding. We have revised the text at line 276.

Result, line 341

Cystamine-stimulated TECs exhibited decreased apoptosis compared to rTGFβ-treated cells.

  1. The authors show a decreased level of apoptosis in hTECs treated with cystamine, however, they do not show evidence of increased apoptosis in the animal models or human samples used in these studies. Did they find differences in TUNEL staining in these samples, for instance?

à According to your comment, we performed a TUNEL assay in the UUO and 5/6 Nx. animal model to show increased apoptosis. We have added the relevant description in the results section.

Results

Apoptosis in UUO kidney showed a continuous increasing over time (0.61 ± 0.16% vs. 3.21 ± 0.69% vs. 8.18 ± 1.32% vs. 58.45 ± 6.54%, Figure 1a, 1d).

Results

The proportion of TUNEL positive apoptosis was also constantly increased at 4 and 8 weeks compared with that of the control group in 5/6 Nx. (0.70 ± 0.23% vs. 16.55 ± 2.17% vs. 77.72 ±4.90%, Figure 1e, 1g).

  1. Please make sure that all acronyms are defined the first time that they are used in the text.

à We have rechecked our manuscript, according to the reviewer’s suggestion.

  1. Please elaborate in the discussion on the limitations of the current study and what are the clinical perspectives of the findings here described. Are any TG2 inhibitors approved for use in humans?

à We have revised the manuscript and have added the limitations and clinical perspectives. Unfortunately, there is no licensed TG2 inhibitor; but there is a TG2 inhibitor investigated in clinical trials. We have added information on the clinical trial in the discussion section. Furthermore, we have revised the conclusion section to describe the clinical perspective of the study findings.

Discussion

Due to the pathologic activities associated with TG2 which result in kidney injury, several investigations have been conducted in this area. Genetic TG2 inhibition in mouse models including UUO and LPS injection exhibited reduced ECM molecule accumulation, de-creased TGFβ1 levels [57], and suppressed infiltration of neutrophils in the kidney [58]. There have been few clinical trial on TG2 inhibitors. A monoclonal Ab for TG2, zampilimab, has been investigated in phase I/II trials for chronic allograft rejections of transplant-ed kidney [59]. Another TG2 inhibitor, ZED1227, ameliorated gluten-induced duodenal mucosal damage in patients with celiac disease. Schuppan et al. have suggested a mechanism that ZED1227 exerts a protective effect by blocking the formation of deamidated gluten and gluten-specific CD4+ T cell activation in the duodenum [60]. However, further investigations are needed to explore the clinical implications of TG2 inhibitors.

This study has several limitations. We demonstrated the anti-apoptotic, anti-fibrotic, and anti-inflammatory effects of inhibiting TG2 only in an in vitro environment. Moreover, the present study revealed only the protective phenomenon caused by TG2 inhibition, and did not include experiments to ascertain the exact mechanisms underlying these protective effects. Further studies using various methods such as multi-omics analysis or in animal experiments are required to investigate the detailed mechanism and the effect of TG2 inhibition in vivo.

Conclusions

With respect to the clinical implications, our study signifies that measuring TG2 expressions could facilitate the evaluation of the severity of CKD progression and presents the detailed beneficial phenomenon that result from pharmacologic TG2 inhibition in kidney TECs.

We thank the reviewers for the helpful comments.

Reviewer 2 Report

The study of Moon et al. demonstrated the high plasma TG2 levels in patients with advanced CKD, and in the animal CKD models. They also investigated the effect of TG2  inhibition in vitro using cystamine, and showed that  TG2 inhibition reduced apoptosis, fibrosis and production of pro-inflammatory IL-8. The results of this study are not entirely new. The antifibrotic effect of pharmacological TG2 inhibition in CKD has been previously demonstrated, as reviewed by Prat-Duran J et al. (Transglutaminase 2 as a novel target in chronic kidney disease - Methods, mechanisms and pharmacological inhibition. Pharmacol Ther. 2021 Jun;222:107787. doi: 10.1016/j.pharmthera.2020.107787). The antiapoptic effect of TG2 inhibition in cultured renal tubular epithelial cells seems to be new observation, although the link between TG2 overexpression in monocytes of graft blood vessels and activation of apoptosis was showed during acute renal allograft injury [Zakrzewicz A. et al.  Monocytic Tissue Transglutaminase in a Rat Model for Reversible Acute Rejection and Chronic Renal Allograft Injury. Mediators Inflamm. 2015;2015:429653. doi: 10.1155/2015/429653].

Moreover, there are some points, which need to be improved.

Fig. 1, g – sirius red positive area should be added

Table 1 and 2. It should be clearly indicated to which values of the patients’ the p value refers to, and whether the values of the analyzed parameters differ between the individual groups of patients with CKD.

 Lines 230-232: „In DM nephropathy, significant glomerular TG2 expression and relatively little TG2 expression were observed” -   opposite, needs explanation

Fig. 2. The association between TG2 and eGFR is only shown on panel d (the values of R and p should be also added). The panels: c, e show TG2 expression in diffrent stages of CKD or in diffrent states of impaired renal function, but there are not associations - please to change the  description of panels c, e.  The same goes for discussion (lines 306-307).

Tab. 2. The caption should be corrected from „Characteristics of patients in the tissue TG2 expression immunohistochemistry group” to „Characteristics of patients with TG2 expression in kidney tissue”

Fig. 3. Panel b – there are really the results of western blot analysis? It seems that there are rather the results of PCR. It is possible to use the same factor (rTGFβ) to study the fibrosis and apoptosis process simultaneously?

Discussion

This section should be shortened and more concentrated on the results of the authors' own research.

- first paragraph – the results should not be repeated, they can be only shortly summarized

- lines 310-322. This paragraph should be discussed with the study of Furini G, et al. Proteomic Profiling Reveals the Transglutaminase-2 Externalization Pathway in Kidneys after Unilateral Ureteric Obstruction. J Am Soc Nephrol. 2018 Mar;29(3):880-905. doi: 10.1681/ASN.2017050479.

Moreover, the other studies concerning TG2 in kidney disease should be cited and discussed, for example: Burhan I et al. Interplay between transglutaminases and heparan sulphate in progressive renal scarring. Sci Rep. 2016 Oct 3;6:31343. doi: 10.1038/srep31343; Da Silva Lodge M, et al. Urinary levels of pro-fibrotic transglutaminase 2 (TG2) may help predict progression of chronic kidney disease. PLoS One. 2022. PMID: 35041708; Prat-Duran J et al. Transglutaminase 2 as a novel target in chronic kidney disease - Methods, mechanisms and pharmacological inhibition. Pharmacol Ther. 2021 Jun;222:107787. doi: 10.1016/j.pharmthera.2020.107787.

- the limitations of the present study should be included, for example that the inhibition of TG2 in cultured TECs can protect these cells from fibrosis, apoptosis and inflammatory response but this has not been confirmed in vivo

- the last sentence „Our study also proposes a potential therapeutic strategy to reduce kidney injury and  fibrosis by inhibiting TG2” seems to be overestimated at this moment.

Author Response

Reviewer 2

The study of Moon et al. demonstrated the high plasma TG2 levels in patients with advanced CKD, and in the animal CKD models. They also investigated the effect of TG2  inhibition in vitro using cystamine, and showed that  TG2 inhibition reduced apoptosis, fibrosis and production of pro-inflammatory IL-8. The results of this study are not entirely new. The antifibrotic effect of pharmacological TG2 inhibition in CKD has been previously demonstrated, as reviewed by Prat-Duran J et al. (Transglutaminase 2 as a novel target in chronic kidney disease - Methods, mechanisms and pharmacological inhibition. Pharmacol Ther. 2021 Jun;222:107787. doi: 10.1016/j.pharmthera.2020.107787). The antiapoptic effect of TG2 inhibition in cultured renal tubular epithelial cells seems to be new observation, although the link between TG2 overexpression in monocytes of graft blood vessels and activation of apoptosis was showed during acute renal allograft injury [Zakrzewicz A. et al.  Monocytic Tissue Transglutaminase in a Rat Model for Reversible Acute Rejection and Chronic Renal Allograft Injury. Mediators Inflamm. 2015;2015:429653. doi: 10.1155/2015/429653].

Moreover, there are some points, which need to be improved.

  1. Fig. 1, g – sirius red positive area should be added

àAccording to reviewer’s recommendation, we added Sirius red stained 5/6 Nx kidney and graph of quantification.

  1. Table 1 and 2. It should be clearly indicated to which values of the patients’ the p value refers to, and whether the values of the analyzed parameters differ between the individual groups of patients with CKD.

à We have revised the tables and have added more description about the tables.

Table 1. Patient characteristics (plasma samples)

Variable

Total

(n = 137)

IgAN

(n = 100)

DMN

(n = 27)

CTL

(n =  10)

P value

Group

IgAN vs. CTL

DMN vs. CTL

IgAN vs. DMN

Age (year)

46.88±15.73

46.09±15.47

53.44±14.06

42.00±5.99

0.041

0.495

0.053

0.019

Sex (male, %)

81 (59.12)

58 (58)

18 (66.66)

5 (50)

0.597

BMI (kg/m2)

24.12±3.56

24.11±3.60

24.51±3.232

23.16±4.11

0.593

0.436

0.301

0.597

CKD stage 1,2 (%)

34 (24.82)

27 (27)

7 (25.93)

CKD stage 3 (%)

63 (45.99)

50 (50)

13 (48.15)

CKD stage 4,5 (%)

30 (21.90)

23 (23)

7 (25.93)

Blood hemoglobin (g/dL)

12.26±2.05

12.41±1.98

11.19±1.95

13.69±1.86

0.005

0.052

0.001

0.005

Serum albumin (g/dL)

3.84±0.54

3.85±0.56

3.69±0.50

4.07±0.39

0.145

0.236

0.039

0.176

Creatinine (mg/dL)

1.76±1.53

1.79±1.59

2.04±1.50

0.78±0.19

0.169

0.048

0.013

0.467

eGFR (CKD-EPI, mL/min/1.73m2)

56.44±33.79

54.08±32.90

47.50±26.42

104.2±22.26

<0.001

<0.001

<0.001

0.340

Urine protein/creatinine ratio     

1.78±2.06

1.55±2.01

3.06±2.08

0.60±0.28

0.002

0.140

0.027

0.001

Plasma TG2 level (pg/mL)

1990±357.8

2030±347.8

1963±334.0

1665±381.3

0.021

0.002

0.001

0.371

Values are means ± SD for continuous variables and n (%) for categorical variables. BMI: Body Mass Index.

Result

The baseline characteristics of patients whose plasma TG2 concentrations were measured are presented in Table 1. The IgAN group demonstrated no significant differences in BMI, serum albumin, serum creatinine, eGFR, and plasma levels compared with those of the DMN group. However, patients with IgAN were younger, had higher blood hemoglobin, and demonstrated lesser urine protein/creatinine ratio when compared to the patients with DMN (Table 1).

Table 2. Characteristics of patients with TG2 expression in kidney tissue

Variable

Total

(n = 94)

IgAN

(n = 56)

DMN

(n = 21)

CTL

(n = 17)

P value

Group

IgAN vs. CTL

DMN vs. CTL

IgAN vs. DMN

Age (year)

44.95±17.90

45.84±15.16

47.24±11.26

39.18±17.90

0.205

0.133

0.099

0.702

Sex (male, %)

56 (56.57)

36 (64.29)

14 (66.67)

6 (35.29)

0.077

BMI (kg/m2)

23.75±3.72

24.28±3.12

25.04±3.97

25.22±4.00

0.694

0.514

0.927

0.476

CKD stage 1,2 (%)

20 (35.71)

5 (23.81)

CKD stage 3 (%)

20 (35.71)

9 (42.86)

CKD stage 4,5 (%)

16 (28.57)

7 (33.33)

Blood hemoglobin (g/dL)

12.26±2.10

12.24±2.00

11.44±2.37

13.34±1.68

0.019

0.043

0.008

0.141

Serum albumin (g/dL)

3.81±0.59

3.85±0.49

3.42±0.77

4.14±0.34

<0.001

0.031

0.001

0.004

Creatinine (mg/dL)

1.56±1.00

1.66±0.99

1.95±1.10

0.75±0.17

<0.001

<0.001

<0.001

0.259

eGFR (CKD-EPI, mL/min/1.73m2)

64.26±37.75

56.49±33.39

50.51±36.81

106.8±19.39

<0.001

<0.001

<0.001

0.498

Urine protein/creatinine ratio

2.42±3.42

1.75±2.17

5.77±5.01

0.47±0.30

<0.001

0.019

<0.001

<0.001

TG2 positive area (%)

8.24±5.61

10.26±5.25

6.76±5.72

3.39±2.24

<0.001

<0.001

0.028

0.013

Values are means ± SD for continuous variables and n (%) for categorical variables. BMI: Body Mass Index

Results

Immunohistochemical staining for TG2 was done to assess the presence of TG2 in kidney biopsies from patients with CKD. Baseline patient characteristics of the biopsy core are represented in Table 2. Tissue donors without CKD were used as the control group. No significant differences were observed in age, BMI, blood hemoglobin, serum creatinine, and eGFR between the IgAN and the DMN groups, although the DMN group showed relatively lower albumin level, smaller TG2 positive area, and more proteinuria (Table 2).

  1. Lines 230-232: „In DM nephropathy, significantglomerular TG2 expression and relatively littleTG2 expression were observed” -   opposite, needs explanation

à According to the reviewer’s comments, we have revised our manuscript. We deleted the text mentioned and have included additional explanation about the results.

In the early stage of IgAN, the tissue expression of TG2 in IgAN was mainly located in the tubule and the interstitial area with minor expression in the glomeruli. Contrarily, in early DMN, TG2 expression was higher than that of IgAN in the glomeruli and was also observed in relatively smaller tubular and interstitial areas. Advanced IgAN showed increased glomerular TG2 expression compared with that in early IgAN, and TG2 expression in advanced DMN was accompanied by tubular and interstitial expression.

  1. Fig. 2. The association between TG2 and eGFR is only shown on panel d (the values of R and p should be also added). The panels: c, e show TG2 expression in diffrent stages of CKD or in diffrent states of impaired renal function, but there are not associations - please to change the  description of panels c, e.  The same goes for discussion (lines 306-307).

à We did not analyze the associations. We have revised the manuscript to state that enhanced TG2 expression was demonstrated in samples from patients with deteriorated kidney function.

Figure 2. TG2 levels were increased with deterioration of kidney function and pathological aggravation in patient samples. (a) The concentration of TG2 was evaluated in the plasma of CKD patients. (b) Representative images of immunohistochemical staining for TG2. (c) Differences in the tissue expression level of TG2 according to kidney function markers, including CKD stage and urine protein/creatinine ratio. (d) Association between TG2 expression and eGFR. (e) Differences in TG2 expression levels according to tubular pathology (*p < 0.05, **p < 0.01, ***p < 0.001).

  1. Tab. 2. The caption should be corrected from „Characteristics of patients in the tissue TG2 expression immunohistochemistry group” to „Characteristics of patients with TG2 expression in kidney tissue”

à We have revised the manuscript according to the reviewer’s comment.

  1. Fig. 3. Panel b – there are really the results of western blot analysis? It seems that there are rather the results of PCR. It is possible to use the same factor (rTGFβ) to study the fibrosis and apoptosis process simultaneously?

à Yes. Figure 3b shows the semiquantified results of western blot analysis. Moreover, rTGFβ treatment can induce fibrotic change and apoptosis simultaneously. Our prior work (Sci Rep. 2019 Sep 17;9(1):13495. doi: 10.1038/s41598-019-49756-z, figure 6) includes experiments similar to that used in the present study.

Discussion

This section should be shortened and more concentrated on the results of the authors' own research.

  1. first paragraph – the results should not be repeated, they can be only shortly summarized

à We agree with your comment. We have summarized the results in the first paragraph.

The present study demonstrated elevation of TG2 expression in the plasma and kidney tissues in advanced CKD compared with early CKD. The inhibition of TG2 activity resulted in decreased apoptosis and attenuation of the release of IL-8 from rTGFβ-stimulated hTECs. Intriguingly, our immunohistochemistry results showed distinguishable tissue TG2 expression patterns between IgAN and DMN, especially in early CKD. Differences in TG2 expression patterns could be clarified through clinical features of both glomerulopathies. Clinically, IgAN showed more hematuria and less proteinuria com-pared with DMN in the early stage. A large amount of proteinuria is more likely to be in-duced by the glomerulus than tubules; in this sense, it is possible to explain the higher TG2 expression in the glomeruli of DMN than IgAN. Our results that patients with more than 1g/day of proteinuria demonstrate higher tissue TG2 expression and plasma TG2 levels also support this. However, despite the differences in the causes of CKD and pattern of TG2 expression, increased immunoreactivity of TG2 was accompanied by worsening of kidney function markers.

  1. lines 310-322. This paragraph should be discussed with the study of Furini G, et al. Proteomic Profiling Reveals the Transglutaminase-2 Externalization Pathway in Kidneys after Unilateral Ureteric Obstruction. J Am Soc Nephrol. 2018 Mar;29(3):880-905. doi: 10.1681/ASN.2017050479.

Moreover, the other studies concerning TG2 in kidney disease should be cited and discussed, for example: Burhan I et al. Interplay between transglutaminases and heparan sulphate in progressive renal scarring. Sci Rep. 2016 Oct 3;6:31343. doi: 10.1038/srep31343; Da Silva Lodge M, et al. Urinary levels of pro-fibrotic transglutaminase 2 (TG2) may help predict progression of chronic kidney disease. PLoS One. 2022. PMID: 35041708; Prat-Duran J et al. Transglutaminase 2 as a novel target in chronic kidney disease - Methods, mechanisms and pharmacological inhibition. Pharmacol Ther. 2021 Jun;222:107787. doi: 10.1016/j.pharmthera.2020.107787.

à We appreciate the helpful comment made by the reviewer. We have included more information on TG2 activity in the introduction and discussion sections.

Introduction

TG2 overexpression has been observed in several kidney disease animal models. In a 5/6 nephrectomy rat model that represents progressive kidney fibrosis, the mRNA and protein expressions of TG2 were observed to be increased. At enhanced levels, TG2 inter-acts with the heparan sulphate family, particularly syndecan-4, which activates TGFβ1 related ECM accumulation [19]. In a kidney transplant animal model, activated monocytes demonstrated TG2 overexpression accompanied with increased levels of the apoptotic marker caspase 3 [20]. Another animal model showed increased urine and serum TG2 concentrations that accompanied kidney disease progression [21].

Discussion

Increased urinary TG2 levels depict allograft inflammation or fibrosis in kidney transplant patients [56], and in CKD patients, urinary TG2 levels could be considered as a potential biomarker for CKD progression [21]. Previous studies have demonstrated the mechanisms underlying TG2 associated-ECM accumulation. TECs excrete TG2 in exosome to interstitial ECM through interaction with syndecan-4, a proteoglycan belonging to the heparan sulfate family [9-10]. In the ECM, TG2 contributes to the crosslinking with integrins and fibronectins that concede resistance to proteolysis, and TG2 inhibition resulted in diminished deposition and facilitated the turnover of ECM components [41-43]. Due to the pathologic activities associated with TG2 which result in kidney injury, several investigations have been conducted in this area. Genetic TG2 inhibition in mouse models including UUO and LPS injection exhibited reduced ECM molecule accumulation, decreased TGFβ1 levels [57], and suppressed infiltration of neutrophils in the kidney [58]. There have been few clinical trial on TG2 inhibitors. A monoclonal Ab for TG2, zampilimab, has been investigated in phase I/II trials for chronic allograft rejections of transplant-ed kidney [59]. Another TG2 inhibitor, ZED1227, ameliorated gluten-induced duodenal mucosal damage in patients with celiac disease. Schuppan et al. have suggested a mechanism that ZED1227 exerts a protective effect by blocking the formation of deamidated gluten and gluten-specific CD4+ T cell activation in the duodenum [60]. However, further investigations are needed to explore the clinical implications of TG2 inhibitors.

  1. the limitations of the present study should be included, for example that the inhibition of TG2 in cultured TECs can protect these cells from fibrosis, apoptosis and inflammatory response but this has not been confirmed in vivo

à As per suggestion, we have added the limitations of our study at the end of discussion section.

This study has several limitations. We demonstrated the anti-apoptotic, anti-fibrotic, and anti-inflammatory effects of inhibiting TG2 only in an in vitro environment. Moreover, the present study revealed only the protective phenomenon caused by TG2 inhibition, and did not include experiments to ascertain the exact mechanisms underlying these protec-tive effects. Further studies using various methods such as multi-omics analysis or in animal experiments are required to investigate the detailed mechanism and the effect of TG2 inhibition in vivo.

  1. the last sentence „Our study also proposes a potential therapeutic strategy to reduce kidney injury and fibrosis by inhibiting TG2” seems to be overestimated at this moment.

à We agree with the reviewer’s comments. Accordingly, we have deleted the last sentence and revised the conclusion section.

Conclusion

Collectively, our work depicts that TG2 expression was increased in various advanced CKD samples. Inhibiting TG2 activity could protect kidney cells from ECM com-ponent accumulation, apoptosis, and inflammatory responses that lead to kidney injury. With respect to the clinical implications, our study signifies that measuring TG2 expressions could facilitate the evaluation of the severity of CKD progression and presents the detailed beneficial phenomenon that result from pharmacologic TG2 inhibition in kidney TECs.

We thank the reviewers for the helpful comments.

Round 2

Reviewer 2 Report

The revised manuscript is suitable for publication.